# The Role of Cryotherapy in Vitreous Concentrations of Topotecan Delivered by Episcleral Hydrogel Implant

**DOI:** 10.3390/pharmaceutics14050903

**Published:** 2022-04-20

**Authors:** Martina Kodetova, Radka Hobzova, Jakub Sirc, Jiri Uhlik, Katerina Dunovska, Karel Svojgr, Ana-Irina Cocarta, Andrea Felsoova, Ondrej Slanar, Martin Sima, Igor Kozak, Pavel Pochop

**Affiliations:** 1Department of Ophthalmology, 2nd Faculty of Medicine, Charles University and Motol University Hospital, 150 06 Prague, Czech Republic; kodetova.martina@gmail.com (M.K.); pavel.pochop@gmail.com (P.P.); 2Institute of Macromolecular Chemistry, Academy of Sciences of the Czech Republic, 162 06 Prague, Czech Republic; hobzova@imc.cas.cz (R.H.); ana_irinac@yahoo.com (A.-I.C.); 3Department of Histology and Embryology, 2nd Faculty of Medicine, Charles University, 150 06 Prague, Czech Republic; jiri.uhlik@lfmotol.cuni.cz (J.U.); andrea.felsoova@lfmotol.cuni.cz (A.F.); 4Department of Medical Chemistry and Clinical Biochemistry, 2nd Faculty of Medicine, Charles University and Motol University Hospital, 150 06 Prague, Czech Republic; katerina.dunovska@fnmotol.cz; 5Department of Pediatric Hematology and Oncology, 2nd Faculty of Medicine, Charles University and Motol University Hospital, 150 06 Prague, Czech Republic; karel.svojgr@fnmotol.cz; 6Clinical and Transplant Pathology Centre, Institute for Clinical and Experimental Medicine, 140 21 Prague, Czech Republic; 7Department of Pharmacology, First Faculty of Medicine, Charles University and General University Hospital in Prague, 128 00 Prague, Czech Republic; ondrej.slanar@lf1.cuni.cz (O.S.); martin.sima@lf1.cuni.cz (M.S.); 8Moorfields Eye Hospital, Abu Dhabi P.O. Box 62807, United Arab Emirates; igor.kozak@moorfields.ae

**Keywords:** hydrogel, HEMA, episcleral implant, topotecan, transscleral diffusion, intraocular delivery, periocular delivery, retinoblastoma, transconjunctival cryotherapy, retina

## Abstract

Transscleral diffusion delivery of chemotherapy is a promising way to reach the vitreal seeds of retinoblastoma, the most common intraocular malignancy in childhood. In this *in vivo* study, the delivery of topotecan via lens-shaped, bi-layered hydrogel implants was combined with transconjunctival cryotherapy to assess whether cryotherapy leads to higher concentrations of topotecan in the vitreous. The study included 18 New Zealand albino rabbits; nine rabbits received a topotecan-loaded implant episclerally and another nine rabbits received transconjunctival cryotherapy superotemporally 2 weeks before implant administration. Median vitreous total topotecan exposures (area under the curve, AUC) were 455 ng·h/mL for the cryotherapy group and 281 ng·h/mL for the non-cryotherapy group, and were significantly higher in the cryotherapy group, similar to maximum levels. Median plasma AUC were 50 ng·h/mL and 34 ng·h/mL for the cryotherapy and non-cryotherapy groups, respectively, with no statistically significant differences between them. In both groups, AUC values in the vitreous were significantly higher than in plasma, with plasma exposure at only approximately 11–12% of the level of vitreous exposure. The results confirmed the important role of the choroidal vessels in the pharmacokinetics of topotecan during transscleral administration and showed a positive effect of cryotherapy on intravitreal penetration, resulting in a significantly higher total exposure in the vitreous.

## 1. Introduction

Retinoblastoma (Rb) is the most common intraocular tumor in children [1]. Advanced intraocular Rb is typically followed by cancer seeding, a type of cell dispersion into an adjacent liquid or semi-liquid compartment. Unfortunately, the presence of seeding is mostly incompatible with successful conservative treatment [2]. Intravitreal chemotherapy has allowed the salvaging of eyes that, until recent years, would have been enucleated. The safety-enhanced technique of intravitreal injection through a tumor-free pars plana site using anti-reflux measures and needle tract sterilization was described in 2012 [3]. The most frequently used drugs for intravitreal injection in the treatment of Rb are melphalan and topotecan (TPT), often administered concomitantly [4,5,6,7]. However, intravitreal chemotherapy with melphalan is associated with retinal toxicity, and therefore loss of retinal function is common [4,8,9]. Compared to melphalan, TPT has a longer intra-ocular half-life [10], and it has been reported that intravitreal monotherapy with TPT shows retinal non-toxicity in a rabbit model [11], even in high doses (up to 50 µg administered weekly) [12]. Recently, the successful control of vitreous disease was achieved with intravitreal TPT applied as a drug solution [13,14,15] or using injectable TPT delivery systems such as hydrogels [16,17] or nanoparticles [18]. Despite these achievements, intravitreal therapy cannot be used in all patients with seeding, and it is even contraindicated in case of uveal or anterior chamber involvement [9,15]. In addition, side effects after intravitreal injections, such as reflux of vitreous at the injection site, retinopathy, vitreous hemorrhage, endophthalmitis, and retinal detachment have been reported [1,19].

Another way to deliver drugs into the posterior eye segment is the periocular approach. Several types of episcleral implants for transscleral diffusion drug administration have been described [20,21,22]. The results of a number of pharmacokinetic studies suggest that conjunctival lymphatic/blood vessels are an important barrier in the delivery of drugs to the vitreous by periocular administration, as the majority of the drug is eliminated before it penetrates the sclera [21,23,24]. Another important barrier to episcleral drug administration that lowers drug levels in the vitreous is the choroidal vasculature [25,26]. Some authors have used cryotherapy to reduce the choroidal blood flow and suppress drug clearance via the choroidal circulation [27,28]. The effects of cryotherapy on ocular tissues have been well described in histological studies [29,30]. The first reports about the role of cryotherapy in the management of Rb were published as early as 1980’ [31,32]. Currently, cryotherapy represents a standard focal treatment modality with few adverse effects [1,33,34]. This treatment destroys small tumors by freezing them at −80 °C and is performed transconjunctivally with a cryoprobe at the tumor site using a triple freeze/thaw technique. Cryotherapy results in a formation of a chorioretinal scar [35]. Additionally, cryoprobes are commonly used in the safety-enhanced technique of intravitreal injection [36,37].

We have previously presented a new bi-layered hydrogel implant for transscleral diffusion delivery for adjunctive therapy in some types of Rb with intravitreal seeding. Our implant is composed of two methacrylate-based layers: an inner hydrophilic drug reservoir made from poly(2-hydroxyethyl methacrylate) (pHEMA), which delivers low molecular weight hydrophilic drugs such as TPT, and an outer hydrophobic layer of poly(2-ethoxyethyl methacrylate) (pEOEMA), which is impermeable and thus suppresses the release of the drug into the surrounding vascularized tissue; the drug is almost exclusively released towards the sclera. *In vitro* properties and *in vivo* proof of concept in a rabbit eye model have been reported [38,39,40].

However, choroidal vessels can still represent a barrier to more effective drug delivery into the Rb-seeded vitreous humor. We hypothesize that an abrogation of choroidal vessels by cryotherapy will lead to higher vitreous TPT levels and lower systemic TPT exposure. In the present study, we used rabbits as an *in vivo* model to study the intravitreal concentration of TPT delivered via the transscleral route by bi-layered hydrogel implants upon co-administration of cryotherapy.

## 2. Materials and Methods

### 2.1. Materials

TPT hydrochloride (≥98%), 2-hydroxyethyl methacrylate (HEMA, ≥99%), ethylene glycol dimethacrylate (EDMA, ≥98%), 2-ethoxyethyl methacrylate (EOEMA, ≥99%), and 2,2´-azobis(2-methylpropionitrile) (AIBN, ≥98) were obtained from Sigma-Aldrich (Prague, Czech Republic). Polypropylene (PP) bars for mold manufacturing were provided by Titan-Multiplast (Smrzovka, Czech Republic). LC-MS-grade methanol, water, and formic acid were obtained from Honeywell (LC-MS grade, Labicom, Czech Republic). The deuterated-labeled internal standards of TPT and TPT hydrochloride were purchased from Toronto Research Chemicals Inc. (North York, ON, Canada).

### 2.2. Hydrogel Implant Preparation and Drug Loading

Preparation: The hydrogel implant is composed of two parts, an inner part made from pHEMA material and outer part of pEOEMA material, which were prepared separately and coupled together before implantation. Preparation of the individual implant parts and their characterization has previously been described in detail [40]. Briefly, the hydrogels were prepared by thermal radical crosslinking polymerization of a mixture of monomers of HEMA or EOEMA, the crosslinker EDMA (0.5 wt% relative to monomer), and initiator the AIBN (0.5 wt% relative to monomer and crosslinker). The polymerization was carried out in molds, which were fabricated by lathe-cutting from polypropylene bars into the shape and dimension that provide lens-shaped hydrogel implants corresponding to the curvature of the rabbit eye. After polymerization, the implants were washed to remove all unreacted low molecular weight and potentially toxic residues from the polymerization. Both the washed and equilibrium-swollen implants were sterilized using a steam sterilizer at 120 °C for 30 min before being used for *in vivo* experiments.

Drug loading: The pHEMA part of the implant was loaded with TPT using the soaking method as follows: equilibrium-swollen pHEMA samples (0.22 g ± 0.08 g) were immersed in 3 mL of 2 mg/mL TPT solution in water and gently stirred at 4 °C for 24 h in the dark. After 24 h, pHEMA samples were removed from the TPT solution, washed three times for 5 s in an excess of water (100 mL) to remove the drops of drug solution from the surface, and then coupled with a sterile pEOEMA cover to prepare the implant for administration onto the rabbit eye.

### 2.3. In Vivo Studies and Sampling Procedures

Eighteen male New Zealand White specified-pathogen-free (SPF) rabbits (Velaz, Prague, Czech Republic) were handled in this study, according to the regulation Nr. 419/2012 of the Ministry of Agriculture of the Czech Republic regarding the use of animals in experiments. Rabbits (weighing 1.7 to 2.2 kg) were anesthetized using intramuscular ketamine hydrochloride (50 mg/kg, Narkamon 100 mg/mL a.u.v. inj., Bioveta Ltd., Ivanovice na Hané, Czech Republic) and xylazine hydrochloride (5 mg/kg, Rometar 20 mg/mL a.u.v. inj., Bioveta Ltd., Ivanovice na Hané, Czech Republic). Half-doses of the same anesthetics were given every hour if prolonged anesthesia was required. Conjunctival sacks were anesthetized by topical oxybuprocaine eye drops (Benoxi 0.4% oph.gtt.sol., Unimed Pharma Ltd., Bratislava, Slovakia) and disinfected with 3 mL of 1% povidone-iodine solution (EGIS Pharmaceuticals Ltd., Budapest, Hungary) before the treatment and sampling.

The rabbits were divided into two groups: a non-cryotherapy (non-cryo) and a cryotherapy (cryo) group. Each animal in the non-cryo group (*n* = 9) received a hydrogel implant episclerally to the right eye, without previous treatment. The animals from the cryo group (*n* = 9) received transconjunctival cryotherapy 14 days before administering the implant to the right eye. Transconjunctival cryotherapy was performed in the superior temporal quadrant at the intended implantation site at 5–6 points for 10–13 s (Erbokryo AE, Erbe Elektromedizin GmbH, Tübingen, Germany). The contralateral eyes were used as controls.

The coupled bi-layered pHEMA/pEOEMA implant was inserted through a conjunctival incision in the superior temporal episcleral zone with the pHEMA part facing the sclera and fixed to the sclera with two Vicryl 6/0 sutures. The implant was placed approximately 1.0–1.5 mm from the limbus and the conjunctiva was sutured again with Vicryl 6/0. The course of the operation was documented by photos, which are presented in Figure 1. Vitreous humor samples (100–200 µL) were obtained from the right eyes using a 25 G needle inserted into the inferior nasal region of the sclera, approximately 3 mm from the limbus at 2, 8, 24 and 48 h, as well as at 7 and 14 days after implant insertion. The vitreous samples from the left eyes were aspirated using a 25 G needle inserted into the superior temporal region of the sclera at 8 h, as a control. These samples were placed in plastic cryotubes, snap frozen in liquid nitrogen, and kept at −80 °C until analysis. Venous blood samples were collected from the marginal ear vein into tubes with ammonium heparinate at 0.5, 1, 2, 8, 24 and 48 h, as well as at 7 and 14 days after hydrogel implant administration. The blood was centrifuged at 1000× *g* and 4 °C for 10 min (Z326K, Hemle Labortechnik, Wehingen, Germany) to isolate the blood plasma. Plasma samples were immediately snap frozen in liquid nitrogen and kept at −80 °C until analysis. Total TPT concentrations were determined by the HPLC-MS/MS method using an Agilent technologies 1290 Infinity II system, including an autosampler, binary pumps, and a thermostated column compartment with 6470 Triple Quad (Agilent technologies, Santa Clara, CA, USA). Processing of plasma and vitreous samples for HPLC-MS/MS analysis and details of the HPLC-MS/MS method are described elsewhere [40].

At each sampling timepoint, the rabbits underwent microscopic and indirect fundoscopic examinations, which were photographically documented. After 14 days of sampling, the animals were euthanatized with an overdose of general anesthesia, and their eyes and retrobulbar tissues were enucleated and fixed with 4% formaldehyde for histopathologic examination. Then, each eyeball was sectioned sagitally, dehydrated using a graded series of ethanol concentrations, cleared using xylene, and embedded in paraffin. Sections of 3 µm thickness were taken and then stained using hematoxylin and eosin. Histopathology findings were evaluated under an Olympus BX43 light microscope (Olympus, Tokyo, Japan), recorded using a DFK 33UX250 camera (Imaging Source, Bremen, Germany), and processed using a NIS Elements AR 5.20.01 image processing software (Laboratory Imaging, Prague, Czech Republic).

### 2.4. Pharmacokinetic and Statistical Analysis

Measured TPT vitreous concentrations were normalized to eyeball volume, while TPT plasma levels were normalized to body weight. Non-detectable TPT levels (levels under the detection limit of 0.5 ng/mL [40]) were treated in the analysis as zero levels. The area under the TPT concentration–time curve from zero to infinity (AUC_0–∞_) was calculated as AUC_0-last_ (up to the last measurable concentration) + AUC_last–∞_ (estimation from the last measured concentration to infinity). AUC_0-last_ was then calculated using the trapezoidal rule, while extrapolation to infinity was estimated as AUC_last–∞_ = C_last_/K_e_, where C_last_ is the last observed quantifiable concentration and K_e_ is the terminal phase elimination rate constant. K_e_ was calculated as the slope of the semi-logarithmic concentration–time profile using linear regression models. Only the last three quantifiable concentrations were used for this purpose.

Descriptive pharmacokinetic parameters were reported as median (interquartile range, IQR) or mean ± standard error of the mean (SEM). The potential difference in parameters between the cryo and non-cryo groups was calculated using the Mann–Whitney test. For comparison of detectable TPT vitreous level up to 48 h after implant administration between the cryo and non-cryo groups, Fisher´s exact test was used. The threshold of statistical significance was considered at *p* ≤ 0.05. All statistical analyses were performed using GraphPad Prism version 8.2.1 (GraphPad Software, Inc., La Jolla, CA, USA).

## 3. Results

### 3.1. Ophthalmic Surgery and Clinical Observations

All cryotherapy procedures were uneventful. Chorioretinal atrophy was seen after 14 days before implant administration by fundoscopic examination, and it was also apparent on histological examination (see Figures below, in Section 3.2).

The lens-shaped pHEMA and pEOEMA layers of the hydrogel implant were characterized by equilibrium swelling in water (38% for pHEMA and 2% for pEOEMA), weight in the swollen state (220 and 200 mg), and central thickness in the swollen state (1.8 and 0.8 mm). The details of these results are described elsewhere [40].

Prior to implantation, the pHEMA layer was loaded with TPT and then glued to the pEOEMA cover layer. The surgical procedure was successful and without any complications, mainly due to the soft and flexible nature of the implant. Throughout the experiment (14 days), rabbits did not display any treatment-induced adverse effects indicating general TPT toxicity, such as weight change or hair loss. Figure 2 shows an example of an eye following implantation and subsequent implant removal. In cryo and non-cryo groups, similar changes in the anterior segment such as conjunctival hyperemia, chemosis, and suffusion due to the surgical manipulation and repeated vitreous samplings were found. Additional ocular changes were assessed 7 and 14 days after implantation and scaled in grades 1 to 4; results obtained together with the description of the grades are presented in Table 1 and Table 2.

### 3.2. Histopathology Findings

The right eyes of rabbits from the cryo group showed the following changes at the cryotherapy site: the sclera was intact, the choroid was thinned, and the retina was partially replaced by choroidal tissue that penetrated the disrupted pigment epithelium (Figure 3A). Only a thin limiting membrane with a small number of cells was found on the inner surface of the defect (Figure 3B). The transitional zone between normal retina/choroid and a chorioretinal scar was observed (Figure 3C). Intact retina and choroid were found outside the cryotherapy site (Figure 3D) and the choroid revealed vascular dilatation in some cases. The ciliary body was usually edematous or showed bleeding into the tissue due to a needle injury or a drop in intraocular pressure during vitreous sampling. Edematous or inflammatory changes of various extents were observed in the conjunctiva. The right eyes from the non-cryo group showed intact eyeball layers (Figure 3E,F). Only one eye revealed corneal vascularization at the limbus and edematous or inflammatory changes were again found in the conjunctiva (Figure 3G,H). The left control eyes did not show any considerable pathological changes in either group.

### 3.3. Pharmacokinetics of TPT

Two rabbits in the cryo group were excluded from the study due to laboratory error (damage of the vitreous samples during processing for HPLC analysis). TPT pharmacokinetic parameters calculated from the vitreous and plasma compartment in both cryo and non-cryo groups are summarized in Table 3. Very low TPT levels (0–1.1 ng/mL) were detected in the vitreous humor of the contralateral eye in both groups (i.e., lower concentrations than those found in the plasma). Detectable vitreous TPT levels up to 48 h after implant administration were observed in 71.4% and 44.4% of subjects in the cryo and non-cryo groups, respectively. The concentration–time profiles showed significantly higher concentration of TPT in vitreous in the cryo group compared to non-cryo group (Figure 4). The concentration in plasma was not significantly different except for C_max_, which was higher for the cryo group at 2 h. In the cryo group, significantly higher total TPT vitreous exposure and TPT maximal vitreous levels were observed (see Table 3). In both groups, AUC_0–∞_ levels were significantly higher in the vitreous than in plasma (*p* = 0.0006 and *p* = 0.0051 in the cryo and non-cryo groups, respectively). Plasma exposure is at only approximately 11–12% of the level of vitreous exposure.

## 4. Discussion

Intravitreal drug delivery using periocular implantation with subsequent transscleral drug diffusion is limited by three main physiological barriers [25,26,41]. The first is the diffusion of the drug molecules through the sclera. The drug diffusion proceeds against the pressure gradient and is driven by the concentration gradient. In the case of sufficient drug concentration provided by the proper drug release kinetics, it is relatively easy to pass. Several papers have demonstrated high transscleral permeability for various molecules [42,43]. This can be even facilitated by selection of a low molecular weight hydrophilic drug such as TPT [38]. The second barrier is the conjunctival blood and lymphatic vasculature. The highly vascularized tissue surrounding the eyeball represents a dynamic barrier that eliminates the drug applied by periocular injection [44,45,46] or released from the administered device [47,48]. The activity of this barrier can be suppressed by using drug-loaded device which is tightly attached to the eyeball and releases the drug entirely towards the side of the sclera [21,39]. The third barrier is the choroidal blood vasculature. It represents a second dynamic barrier that decreases the drug concentration after its penetration through the sclera. This barrier can be suppressed by use of vasoconstrictors to temporarily limit the blood flow [20,49] or by physical treatment such as cryotherapy, which necrotizes the choroid, minimizes blood and lymph transport, and consequently suppresses drug clearance [27,28].

In the present work, we have combined two strategies to enhance the intravitreal TPT penetration. We used a lens-shaped, bi-layered hydrogel implant that releases TPT entirely towards the sclera together with transconjunctival cryotherapy to eliminate TPT clearance via the choroidal blood vessels.

Macroscopic ophthalmological observations revealed no difference between the cryo and non-cryo groups of rabbits. In both groups, similar post-surgical changes were found in both the anterior and posterior segments, apparently caused by the surgical manipulation and vitreous sampling. Defects in the conjunctiva covering the implant and implant dislocation on the surface of the cornea were observed in both groups, which was probably due to the loosening of the absorbable suture. In the cryo group, the implant dislocated earlier and more frequently. The hypothesis that the conjunctiva can be more fragile shortly after cryotherapy has not been histologically confirmed. Dislocation of the implant, however, did not caused any irreversible impact on the ocular tissues. Moreover, it occurred at a time when no TPT has already being delivered. Shorter implantation times may be considered to avoid this dislocation in the future.

Histological findings confirmed the expected chorioretinal atrophy following cryotherapy. Outside the cryotherapy site, the choroid and retina remained intact. The extent of edematous and inflammatory changes in the conjunctiva was found to be similar in both groups. Two weeks after cryotherapy, a chorioretinal scar was observed, which is in accordance with other reports. Steel et al. observed clinical evidence of a chorioretinal scar within 1–4 weeks following cryotherapy [30]. Robinson et al. suggested a minimum of 1 month to form a mature chorioretinal scar [27]. Our experiments showed that a 2-week period between cryotherapy and implant administration is sufficient to achieve the intended cryotherapy effect.

The results of the pharmacokinetic evaluation suggest the following conclusions. The plasma levels with C_max_ values at 2 h 6.0 ng/mL and total TPT plasma exposure (AUC_0–∞_) with median values of 50.3 for the cryo group and 33.8 ng·h/mL for non-cryo group, suggest that the pEOEMA coating prevents the TPT release into the surrounding tissue and therefore the drug clearance to the blood circulation prior to sclera penetration does not occur. Although the C_max_ in plasma was determined to be significantly higher for the cryo group than for the non-cryo group, it was just a concentration determined in one time interval, in other time intervals the difference was not significant. Similarly, the AUC levels did not differ significantly. The total TPT vitreous exposure was approximately 1.6 times higher in the cryo group versus the non-cryo group.

This statistically significant difference together with the significantly higher TPT maximal vitreous levels observed suggest that the cryo group is likely to have slower drug elimination from the vitreous compartment. This supports the positive effect of cryotherapy on increased drug delivery to the vitreous.

We acknowledge a relatively low number of experimental animals as a limitation. Together with multiple vitreous samplings, this provides a relatively high level of variability in the pharmacokinetic data. Experimental arrangements with collection of one sample from each eye can improve this variability, bring more accurate results of pharmacokinetic parameters [11], and will be the subject of further studies.

It is important to note that our rabbit model does not completely resemble the human situation. Besides the anatomical differences between the rabbit and human eyes, in non-tumor bearing animals, the blood–retinal barrier is intact, while in the Rb-affected eye, this barrier is altered [50]. This may have an effect on the penetration of episclerally delivered chemotherapy into the vitreous. Therefore, our further work will focus on an Rb animal model that resembles the situation in childhood Rb more closely.

## 5. Conclusions

The presented work assesses the contribution of the transconjunctival cryotherapy on the pharmacokinetics of the TPT delivered via bi-layered hydrogel episcleral implants. The results demonstrate low plasma exposures when compared to vitreal exposures and significantly higher total TPT exposures in vitreous in rabbit group with prior cryotherapy. The observations confirm the important role of the choroidal vessels in the pharmacokinetics of TPT during transscleral administration and show an enhanced effect of following cryodestruction of the choroidal vessels leading to significantly higher TPT concentrations in the vitreous. For our future experiments, we will use cryotherapy prior to transscleral administration of chemotherapy as the standard of care.

## Figures and Tables

**Figure 1 pharmaceutics-14-00903-f001:**
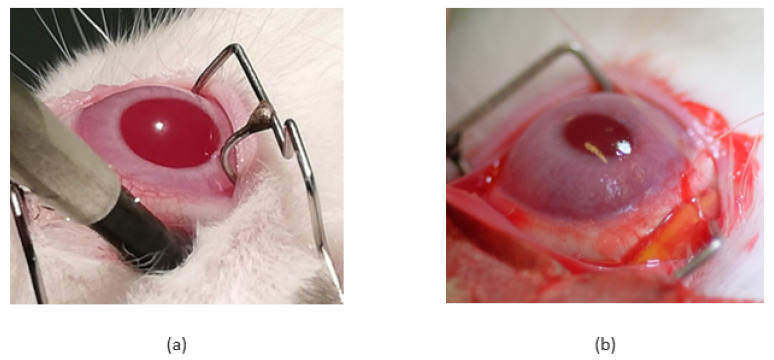
(**a**) Transconjunctival cryotherapy, (**b**) The appearance of the eye with the episclerally fixed implant before conjunctival closure.

**Figure 2 pharmaceutics-14-00903-f002:**
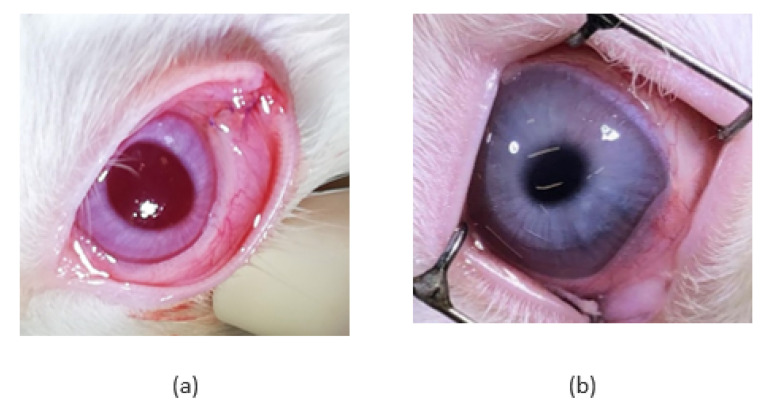
The appearance of the eye in the non-cryo group: (**a**) immediately after the implant administration and (**b**) 2 weeks later, after the implant removal.

**Figure 3 pharmaceutics-14-00903-f003:**
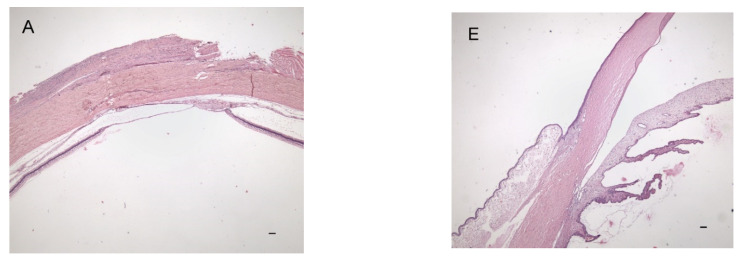
Histological findings: cryo group (**A**–**D**) and non-cryo group (**E**–**H**). Changes at the cryotherapy site: (**A**) intact sclera, thinned choroid, and retina partially replaced by choroidal tissue that has penetrated the disrupted pigment epithelium, (**B**) thin limiting membrane with few cells on the surface (arrows), (**C**) The transitional zone border between normal retina/choroid and a chorioretinal scar, (**D**) Intact retina and choroid outside the site of applied cryotherapy, (**E**,**F**) intact eyeball layers, and (**G**,**H**) edematous and inflammatory changes in the conjunctiva. Bars = 100 µm.

**Figure 4 pharmaceutics-14-00903-f004:**
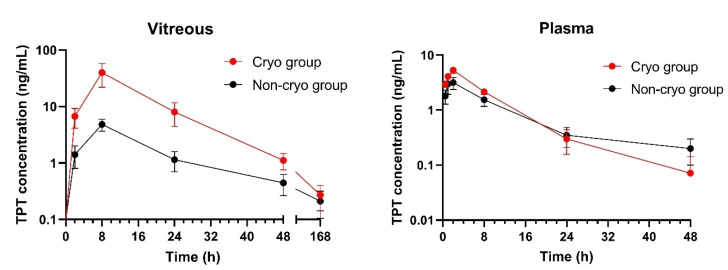
Vitreous and plasma TPT concentration–time profiles in the cryo and non-cryo group. Data are expressed as means ± SEM.

**Table 1 pharmaceutics-14-00903-t001:** Clinical grading of findings observed 7 days after implantation.

	Non-Cryo Group	Cryo Group
Corneal haze ^a^		
Grade 1	1 (11%)	-
Grade 2	-	1 (11%)
Grade 3	-	1 (11%)
Grade 4	-	-
Corneal vascularization approximately 2 mm beyond the limbus in the upper quadrants	1 (11%)	-
Implant uncovered with conjunctiva ^b^		
Grade 1	-	-
Grade 2	1 (11%)	1 (11%)
Grade 3	-	-
Implant dislocated on the surface of the cornea and covers the entire cornea	-	2 (22%)

Grading scales ^a^ Corneal haze: *1 Focal haze and the iris can be seen; 2 Diffuse haze and the iris can be seen; 3 Cloudy cornea, no iris details, and the pupil can be seen; 4 Opaque cornea, no iris details, and poor pupillary details.*
^b^
*Implant uncovered with conjunctiva: 1 Defect in conjunctiva <2 mm; 2 Defect in conjunctiva >2 mm; 3 Implant completely uncovered with conjunctiva.*

**Table 2 pharmaceutics-14-00903-t002:** Clinical grading of findings observed 14 days after implantation (the day of enucleation).

	Non-Cryo Group	Cryo Group
Corneal haze ^a^		
Grade 1	2 (22%)	-
Grade 2	-	-
Grade 3	-	2 (22%)
Grade 4	-	-
Corneal vascularization over the upper limbus ^b^		
Grade 1	2 (22%)	1 (11%)
Grade 2	3 (33%)	4 (44%)
Grade 3	-	-
Vitreous hemorrhage ^c^		
Grade 1	1 (11%)	-
Grade 2	1 (11%)	-
Grade 3	2(22%)	1 (11%)
Implant uncovered with conjunctiva ^d^		
Grade 1	-	-
Grade 2	1 (11%)	2 (22%)
Grade 3	-	-
Implant dislocated on the surface of the cornea and covers the entire cornea	4 (44%)	5 (56%)

Grading scales ^a^ Corneal haze: *1 Focal haze and the iris can be seen; 2 Diffuse haze and the iris can be seen; 3 Cloudy cornea, no iris details, and the pupil can be seen; 4 Opaque cornea, no iris details, and poor pupillary details.*
^b^
*Corneal vascularization: 1 Clear cornea with peripheral corneal vascularization of <2 mm; 2 Peripheral corneal vascularization of >2 mm, sparing the central cornea; 3 Corneal vascularization involving the central cornea.*
^c^
*Vitreous hemorrhage: 1 Mild (not preventing detailed fundus examination); 2 Moderate (obscuring at least one or two quadrants of retinal detail); 3 Severe (too dense for optic disk visualization).*^d^
*Implant uncovered with conjunctiva: 1 Defect in conjunctiva <2 mm; 2 Defect in conjunctiva >2 mm; 3 Implant completely uncovered with conjunctiva.*

**Table 3 pharmaceutics-14-00903-t003:** TPT pharmacokinetic parameters calculated from vitreous and plasma compartment in the cryo and non-cryo groups.

	Pharmacokinetic Parameter	Cryo Group	Non-Cryo Group	*p*-Value
Vitreous	AUC_0–∞_ (ng·h/mL)	454.6 (291.9–1260.0)	281.4 (180.0–321.3)	0.0480
	C_max_ (ng/mL)	20.6 (6.1–63.5)	2.8 (2.2–8.2)	0.0073
	T_max_ (h)	8 (8–8)	8 (8–8)	>0.9999
Plasma	AUC_0–∞_ (ng·h/mL)	50.3 (44.3–70.6)	33.8 (19.0–91.1)	0.7104
	C_max_ (ng/mL)	6.0 (3.2–6.5)	2.2 (1.4–4.8)	0.0210
	T_max_ (h)	2 (1.5–2)	2 (2–2)	0.1125
Vitreous/Plasma Ratio	AUC_0–∞_	8.7 (4.7–30.7)	7.9 (4.2–17.6)	0.8763
	C_max_	4.3 (2.3–10.6)	1.4 (0.9–3.1)	0.0549

Data are expressed as median (IQR).

## Data Availability

All data related to this work are available on request from the corresponding author.

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
