# Peer review of "The Role of Cryotherapy in Vitreous Concentrations of Topotecan Delivered by Episcleral Hydrogel Implant"

_pharmaceutics, 2022, doi:10.3390/pharmaceutics14050903_

Round 1

Reviewer 1 Report

The subject of this study is very interesting, it is well organized even if the technique is rather invasive.

The results could be a little more articulated, for example the clinical findings could be reported by giving a score to the different manifestations by dividing them into mild, moderate and intense and not only there or not.

The discussion is very limited, this work deserves a more in-depth debate

Author Response

Reviewer 1

English language and style

( ) Extensive editing of English language and style required
( ) Moderate English changes required
(x) English language and style are fine/minor spell check required
( ) I don't feel qualified to judge about the English language and style

Yes

Can be improved

Must be improved

Not applicable

Does the introduction provide sufficient background and include all relevant references?

(x)

( )

( )

( )

Is the research design appropriate?

(x)

( )

( )

( )

Are the methods adequately described?

(x)

( )

( )

( )

Are the results clearly presented?

( )

(x)

( )

( )

Are the conclusions supported by the results?

( )

(x)

( )

( )

Comments and Suggestions for Authors

The subject of this study is very interesting, it is well organized even if the technique is rather invasive.

The results could be a little more articulated, for example the clinical findings could be reported by giving a score to the different manifestations by dividing them into mild, moderate and intense and not only there or not.

We appreciate the Reviewer's comment, using the scored evaluation of clinical findings will certainly improve the level of our manuscript. In this regard, we added grading 1 to 4 with description to Table 1 and 2.

The discussion is very limited, this work deserves a more in-depth debate

We agree with the Reviewer that more detailed conclusions and statements would be beneficial, in this regard several particular changes in the “Discussion” has been done. For making more deep discussion, further extensive experimental data will be required, e.g. the comparison of pharmacokinetics of the drug delivered with bi-layered hydrogel implant alone, and together with applied cryotherapy and coadministration of vasoconstrictor would be certainly interesting. The aim of the current submitted manuscript is the description of the cryotherapy contribution to the topotecan delivery via episcleral hydrogel device in principle. We believe that releasing of the results in such extent is beneficial for other authors dealing with this drug delivery route.

Reviewer 2 Report

My comment and question are as follows,

  1. This paper has a clear purpose that is practically applicable to clinical practice, and it is an excellent study logically described under well-designed methods. However, as the authors stated in the limitation, the number of N is too small, and the result itself may has weak evidence due to repeated sampling (especially in vitreous).
  2. Introduction is too lengthy. As this study has a clear purpose, it would be good to briefly summarize some key points that are important for logical development.
  3. What do you think it means to have a significantly higher Cmax of Plasma in Cryo group?
  4. In addition to Choroidal vessels, Cryo may have a short-term impact on conjunctival vessels and sclera. Could these effects affect drug penetration? Or should it be considered that the effect on this is negligible after about 14 days after cryo?

Author Response

Reviewer 2

English language and style

( ) Extensive editing of English language and style required
( ) Moderate English changes required
( ) English language and style are fine/minor spell check required
(x) I don't feel qualified to judge about the English language and style

Yes

Can be improved

Must be improved

Not applicable

Does the introduction provide sufficient background and include all relevant references?

(x)

( )

( )

( )

Is the research design appropriate?

(x)

( )

( )

( )

Are the methods adequately described?

(x)

( )

( )

( )

Are the results clearly presented?

(x)

( )

( )

( )

Are the conclusions supported by the results?

(x)

( )

( )

( )

Comments and Suggestions for Authors

My comment and question are as follows,

  1. This paper has a clear purpose that is practically applicable to clinical practice, and it is an excellent study logically described under well-designed methods. However, as the authors stated in the limitation, the number of N is too small, and the result itself may has weak evidence due to repeated sampling (especially in vitreous).

We fully agree with the Reviewer that the small number of animals and repeated vitreous sampling represent a limitation and may affect the results to some extent. However, we believe that despite this limitation, we have clearly demonstrated that our strategy of using episcleral implants in combination with cryotherapy leads to drug vitreous concentrations at the therapeutically promising levels while maintaining the overall exposition low. Within the future work, we plan to rearrange experiments with an emphasis on animal sampling only once to obtain more relevant data for pharmacokinetic analysis. This is mentioned in the “Discussion” section in the last two paragraphs.

  1. Introduction is too lengthy. As this study has a clear purpose, it would be good to briefly summarize some key points that are important for logical development.

We appreciate the Reviewer's comment. Several changes in the “Introduction” part have been made in this regard.

  1. What do you think it means to have a significantly higher Cmax of Plasma in Cryo group?

We thank Reviewer for this comment. Indeed, Cmax in plasma was significantly higher for Cryo group, however, AUC, which is more relevant parameter to be observed, was not significantly higher. We suppose that this is only by chance finding as the Cmax, unlike AUC, is determined by a single concentration point. Thus, it is possible that the Cmax may not have been accurately captured with the sampling time scheme used. The changes in manuscript have been done on page 12 in chapter 3.3 and page 15 in discussion part. We also added the concentration-time profiles and it is apparent that the significant difference was just in one time interval.

  1. In addition to Choroidal vessels, Cryo may have a short-term impact on conjunctival vessels and sclera. Could these effects affect drug penetration? Or should it be considered that the effect on this is negligible after about 14 days after cryo?

We thank Reviewer for this comment. While the sclera is largely avascular, the episclera and conjunctiva contain vascular networks. The cryotherapy is commonly used in the surgical treatment of retinal detachments and has not been observed to cause any clinical damage to these vascular networks. Simultaneously with transscleral cryotherapy, subconjunctival injection of drugs (such as antibiotics) is administered without any compromise in their efficacy as a result of cryotherapy. As such, the effect of cryotherapy on conjunctival and episcleral vessels is considered clinically negligible.

Reviewer 3 Report

The paper evaluated the effect of cryotherapy on transscleral delivery of topotecan (TPT) from an episcleral hydrogel implant.  The experiments were conducted in rabbits in vivo for 14 days for pharmacokinetics and histology evaluations.  The results indicate higher vitreous concentration of TPT delivered from the implant for the animals with cryotherapy.  The results are interesting and useful for the research community but the utility of cryotherapy to enhance transscleral drug delivery in clinical practice is uncertain.  The following are the point-by-point comments.

  1. It is mentioned in the paper that “detectable TPT vitreous levels . . . were observed in 71.4% and 44.4% of subjects. . . .” How was the undetectable TPT data point treated in the analyses such as in the calculations of the AUC and Cmax in Table 3?  They should not be excluded in the analyses.

  1. The pharmacokinetics profiles are not presented. The data are summarized in a table only.  These “concentration vs. time” profiles must be provided for the readers to interpret the data (in addition to the summary in the table). 

  1. Some statements are confusing and difficult to follow. Some of them are related to the use of terminologies, choices of words, style of writing, and the use of English.

  1. The sentence in line 312 requires clarification. Cryotherapy can reduce the effect of the dynamic barrier for drug delivery.  There are no data to support that cryotherapy lead to lower elimination rate from the vitreous. 

  1. The sentence in line 322 requires clarification. The blood-retinal barrier is related to systemic delivery.  It is unclear how a disrupted blood-retinal barrier will affect transscleral delivery in a positive way.

  1. More descriptions and references on the dynamic barrier (clearance by blood vasculature and lymph) for transscleral delivery can be included in the Introduction and/or Discussion.

  1. There is inconsistency in the organization in Section 3 of the paper: a subsection (first paragraph) under Section 3 without any subheading.

Author Response

English language and style

( ) Extensive editing of English language and style required
(x) Moderate English changes required
( ) English language and style are fine/minor spell check required
( ) I don't feel qualified to judge about the English language and style

Yes

Can be improved

Must be improved

Not applicable

Does the introduction provide sufficient background and include all relevant references?

( )

(x)

( )

( )

Is the research design appropriate?

(x)

( )

( )

( )

Are the methods adequately described?

( )

(x)

( )

( )

Are the results clearly presented?

( )

( )

(x)

( )

Are the conclusions supported by the results?

(x)

( )

( )

( )

Comments and Suggestions for Authors

The paper evaluated the effect of cryotherapy on transscleral delivery of topotecan (TPT) from an episcleral hydrogel implant. The experiments were conducted in rabbits in vivo for 14 days for pharmacokinetics and histology evaluations. The results indicate higher vitreous concentration of TPT delivered from the implant for the animals with cryotherapy. The results are interesting and useful for the research community but the utility of cryotherapy to enhance transscleral drug delivery in clinical practice is uncertain. The following are the point-by-point comments.

  1. It is mentioned in the paper that “detectable TPT vitreous levels . . . were observed in 71.4% and 44.4% of subjects. . . .” How was the undetectable TPT data point treated in the analyses such as in the calculations of the AUC and Cmax in Table 3?  They should not be excluded in the analyses.

We thank to reviewer for this comment, it was not clearly described in the manuscript. Undetectable TPT levels were those under the detection limit of the HPCL-MS/MS method used, i.e. below 0.5 ng/ml. These values were not excluded from the analysis, they were treated as zero levels. We have added this note to paragraph 2.4.

  1. The pharmacokinetics profiles are not presented. The data are summarized in a table only. These “concentration vs. time” profiles must be provided for the readers to interpret the data (in addition to the summary in the table).

We agree with the reviewer, the “concentration vs. time” profiles were added as Figure 4.

  1. Some statements are confusing and difficult to follow. Some of them are related to the use of terminologies, choices of words, style of writing, and the use of English.

With the whole respect to the reviewer, the topic is multidisciplinary, each specialist uses the terminology which is common in his field of scientific interest, so it is difficult to change. We went through the whole document and rewrote some sentences to improve the language and statement level. In addition, the manuscript was proofread by Proof-reading-service company.

  1. The sentence in line 312 requires clarification. Cryotherapy can reduce the effect of the dynamic barrier for drug delivery.  There are no data to support that cryotherapy lead to lower elimination rate from the vitreous.

We agree with the Reviewer, that there is no information of direct effect of cryotherapy itself to lower elimination of drug from vitreous. However, higher TPT Cmax and AUC in the vitreous suggest that cryo group is likely to have slower drug elimination. The sentence was rewritten in this regard.

  1. The sentence in line 322 requires clarification. The blood-retinal barrier is related to systemic delivery.  It is unclear how a disrupted blood-retinal barrier will affect transscleral delivery in a positive way.

We agree with the Reviewer, it is unclear whether the effect of changes on BRB related with Rb will have a positive effect on the episcleral drug delivery. However, it can be supposed, that BRB in Rb affected eye is altered compared to healthy eye and the pharmacokinetics can thus differ. The statement was reformulated and reference relating this issue was added (Reference No 50, Wang et al 2020).

  1. More descriptions and references on the dynamic barrier (clearance by blood vasculature and lymph) for transscleral delivery can be included in the Introduction and/or Discussion.

We appreciate the Reviewer's comment. We have added relevant references and we have extended this issue in “Discussion” part. We have discussed “the outer” dynamic barrier, i.e. drug clearance via conjunctival lymphatic/blood vessels, in our previous paper introducing bi-layered implant as this barrier can be suppressed by using implants with unidirectional release. The main purpose of this study was to assess the role of applied cryotherapy in the clearance of drug via choroidal blood flow, and therefore more emphasis was placed on this dynamic ocular barrier.

  1. There is inconsistency in the organization in Section 3 of the paper: a subsection (first paragraph) under Section 3 without any subheading.

We thank to reviewer for this comment, we added subheading for the first subsection “3.1. Ophthalmic surgery and clinical observations” and the others we have accordingly renumbered.

Reviewer 4 Report

The manuscript is well written and graphically presented.

Some minor comments:

The authors claimed that the drug is the second most frequently used intravitreally. This is not correct. corticosteroids and anti VEGF are the most widely used. so please rephrase.

The introduction should highlight previous work has been done and published on topotecan.

Conclusion section needs to be expanded a bit to highlight key research findings.

Author Response

Reviewer 4

English language and style

( ) Extensive editing of English language and style required
( ) Moderate English changes required
( ) English language and style are fine/minor spell check required
(x) I don't feel qualified to judge about the English language and style

Yes

Can be improved

Must be improved

Not applicable

Does the introduction provide sufficient background and include all relevant references?

(x)

( )

( )

( )

Is the research design appropriate?

(x)

( )

( )

( )

Are the methods adequately described?

(x)

( )

( )

( )

Are the results clearly presented?

(x)

( )

( )

( )

Are the conclusions supported by the results?

(x)

( )

( )

( )

Comments and Suggestions for Authors

The manuscript is well written and graphically presented.

Some minor comments:

The authors claimed that the drug is the second most frequently used intravitreally. This is not correct. corticosteroids and anti VEGF are the most widely used. so please rephrase.

We agree with the reviewer, we had in mind that TPT is the second most frequently used intravitreal injected drug in the treatment of retinoblastoma. In this regard, we have rephrased the text.

The introduction should highlight previous work has been done and published on topotecan.

We thank Reviewer for this comment. In this regard, we have rewritten the “Introduction” part and added newer relevant citations.

  • Del Sole, M. J., Clausse, M., Nejamkin, P., Cancela, B., Del Río, M., Lamas, G., Lubieniecki, F., Francis, J. H., Abramson, D. H., Chantada, G., & Schaiquevich, P. (2022). Ocular and systemic toxicity of high-dose intravitreal topotecan in rabbits: Implications for retinoblastoma treatment. Experimental Eye Research, 218(March), 109026. https://doi.org/10.1016/j.exer.2022.109026
  • Koç, I., Kiratli, H., & Chawla, B. (2021). Update on Intravitreal Chemotherapy for Retinoblastoma. Advances in Ophthalmology and Optometry, 6, 101–118. https://doi.org/10.1016/j.yaoo.2021.04.008

Conclusion section needs to be expanded a bit to highlight key research findings.

We appreciate the Reviewer's comment. The “Conclusion” has been extended and rewritten to better reflect our key results achieved within this study.

Round 2

Reviewer 1 Report

Accepted

Author Response

Dear Reviewer,

The manuscript has been proofread by a professional English language editor from Proof-reading services company. The author's team has carefully checked the text and made several stylistic, language and spelling changes. The changes made are visible using "Track Changes" function.

Reviewer 3 Report

There are still some minor issues with terminologies and English language.  Please check and proofread.

Author Response

(The authors gave the same response as above.)
